# Impact of Preoperative Chemotherapy Features on Patient Outcomes after Hepatectomy for Initially Unresectable Colorectal Cancer Liver Metastases: A LiverMetSurvey Analysis

**DOI:** 10.3390/cancers14174340

**Published:** 2022-09-05

**Authors:** Pasquale F. Innominato, Valérie Cailliez, Marc-Antoine Allard, Santiago Lopez-Ben, Alessandro Ferrero, Hugo Marques, Catherine Hubert, Felice Giuliante, Fernando Pereira, Esteban Cugat, Darius F. Mirza, Jose Costa-Maia, Alejandro Serrablo, Real Lapointe, Cristina Dopazo, Jose Tralhao, Gernot Kaiser, Jinn-Shiun Chen, Francisco Garcia-Borobia, Jean-Marc Regimbeau, Oleg Skipenko, Jen-Kou Lin, Christophe Laurent, Enrico Opocher, Yuichi Goto, Benoist Chibaudel, Aimery de Gramont, René Adam

**Affiliations:** 1Oncology Department, Ysbyty Gwynedd, Betsi Cadwaladr University Health Board, Bangor LL57 2PW, UK; 2Cancer Research Centre, Warwick Medical School, University of Warwick, Coventry CV4 7AL, UK; 3UPR “Chronotherapy, Cancers and Transplantation”, Faculty of Medicine, Paris-Saclay University, 94800 Villejuif, France; 4AP-HP, Hôpital Paul Brousse, Centre Hépato-Biliaire, Research Unit Chronotherapy, Cancers and Transplantation, University Paris Saclay, 94800 Villejuif, France; 5Hospital Josep Trueta, 17007 Girona, Spain; 6Ospedale Mauriziano Umberto I, 10128 Torino, Italy; 7Hospital de Curry Cabral, 1050-099 Lisboa, Portugal; 8Cliniques Universitaires Saint-Luc, 1200 Bruxelles, Belgium; 9School of Medicine, Catholic University, 00168 Rome, Italy; 10Hospital De Fuenlabrada, 28942 Madrid, Spain; 11Hospital Mutua de Terrassa, 08221 Terrassa, Spain; 12Queen Elizabeth Hospital Birmingham, Birmingham B15 2GW, UK; 13Hospital de Sao Joao, 4200-319 Porto, Portugal; 14Miguel Servet University Hospital, 50009 Zaragoza, Spain; 15CHUM, Hôpital Saint-Luc, Montreal, QC H2X 3E4, Canada; 16Vall d’Hebron University Hospital, 08035 Barcelona, Spain; 17Centro Hospitalar e Universitario de Coimbra, 3004-561 Coimbra, Portugal; 18Essen University Hospital, 45147 Essen, Germany; 19Chang Gung Memorial Hospital, Tao-Yuan 33305, Taiwan; 20Consorci Hospitalari Parc Tauli, 08208 Sabadell, Spain; 21CHU d’Amiens Picardie-Site Nord, 80080 Amiens, France; 22National Research Center of Surgery, 119991 Moscow, Russia; 23Taipei Veterans General Hospital, Taipei 11217, Taiwan; 24Hôpital Saint-André, 33000 Bordeaux, France; 25Ambulatorio di Chirurgia Epato-Biliare, Ospedale San Paolo, 20142 Milano, Italy; 26Department of Surgery, School of Medicine, Kurume University, Fukuoka 830-0011, Japan; 27Oncology Department, Hôpital Franco-Britannique–Fondation Cognacq-Jay, 92300 Levallois-Perret, France

**Keywords:** colorectal cancer, liver metastases, hepatectomy, neo-adjuvant chemotherapy, preoperative chemotherapy, onco-surgical approach, liver resection, LiverMetSurvey, real-world evidence, downsizing, irresectable hepatic metastases

## Abstract

**Simple Summary:**

The features of preoperative systemic anticancer therapy associated with best outcomes after resection of initially-irresectable liver metastases from colorectal cancer are yet to be identified. We harnessed data from a prospective international surgical database (LiverMetSurvey) to explore the duration and modalities of preoperative systemic anticancer therapy associated with longer overall survival in this clinical setting. Our study included 2793 patients having undergone liver surgery after preoperative systemic anticancer treatment for initially irresectable disease. We found that short (<7 or <13 cycles in 1st or 2nd line, respectively) duration was associated with longer survival outcomes, independently from other prognostic factors. Conversely, all the comparisons between different conventional active regimens displayed similar results. Our findings support the recommended onco-surgical approach of aiming at performing liver surgery as soon as technically feasible after response to preoperative systemic anticancer therapy in hepatic metastases from colorectal cancer, initially not amenable to surgery. The results of this study also suggest that, provided the systemic anticancer therapy regimen is active, the choice of the drugs used bears overall little if any impact on the outcomes.

**Abstract:**

**Background:** Prognostic factors have been extensively reported after resection of colorectal liver metastases (CLM); however, specific analyses of the impact of preoperative systemic anticancer therapy (PO-SACT) features on outcomes is lacking. **Methods:** For this real-world evidence study, we used prospectively collected data within the international surgical LiverMetSurvey database from all patients with initially-irresectable CLM. The main outcome was Overall Survival (OS) after surgery. Disease-free (DFS) and hepatic-specific relapse-free survival (HS-RFS) were secondary outcomes. PO-SACT features included duration (cumulative number of cycles), choice of the cytotoxic backbone (oxaliplatin- or irinotecan-based), fluoropyrimidine (infusional or oral) and addition or not of targeted monoclonal antibodies (anti-EGFR or anti-VEGF). **Results:** A total of 2793 patients in the database had received PO-SACT for initially irresectable diseases. Short (<7 or <13 cycles in 1st or 2nd line) PO-SACT duration was independently associated with longer OS (HR: 0.85 *p* = 0.046), DFS (HR: 0.81; *p* = 0.016) and HS-RFS (HR: 0.80; *p* = 0.05). All other PO-SACT features yielded basically comparable results. **Conclusions:** In this international cohort, provided that PO-SACT allowed conversion to resectability in initially irresectable CLM, surgery performed as soon as technically feasible resulted in the best outcomes. When resection was achieved, our findings indicate that the choice of PO-SACT regimen had a marginal if any, impact on outcomes.

## 1. Introduction

Colorectal cancer remains the third cancer in terms of incidence and mortality in both sexes [1]. Liver involvement at diagnosis or as a recurrence occurs in more than 50% of patients [2]. For them, surgical resection of liver metastases remains the only chance of prolonged complete remission, with observed 5-year survival rates ranging from 25% to 60% [3,4]. Conversely, 5-year overall survival (OS) rate is markedly worse for patients with inoperable disease, reaching 5% at best [5,6]. Therefore, an international panel of multidisciplinary experts developed recommendations for patients with liver metastases, stating that the onco-surgical treatment strategy should be directed toward resectability [7,8]. Thus, active systemic anticancer treatment has been demonstrated to be able to convert about 30% of patients with the initially-unresectable disease to liver surgery [9,10]; however, there is no universal consensus or definitive dedicated randomized-controlled trials regarding the optimal chemotherapy features in terms of long-term outcomes [4,11,12]. Indeed, although incremental improvements in response rates and overall survival have been achieved with optimization of systemic anticancer treatment in metastatic disease, for instance with targeted anti-EGFR agents [6], and promising evidence as conversion chemotherapy has been observed [13], surprising results have appeared recently when used in the peri-operative setting of resectable hepatic disease [14].

In this context of uncertainty regarding the impact of the type and modalities of pre-operative systemic anticancer therapy (PO-SACT) on patients’ outcomes and aware of the clinical relevance of real-world data [15], we harnessed data from the prospective international cohort LiverMetSurvey [16,17], in order to determine the PO-SACT features associated with the longest overall survival of resected patients with initially-unresectable liver metastases. The identification of prognostic characteristics of PO-SACT would contribute to improving tailored schedule selection with the best outcomes.

## 2. Patients and Methods

### 2.1. LiverMetSurvey and Patient Selection

LiverMetSurvey is an international, internet-based registry designed to assess the efficacy of multimodality treatment options for colorectal liver metastases (CLM) [16,17]. From January 1995 (and from January 2005, prospectively) to December 2020, 27,210 consecutive patients operated in 280 centers from 54 countries had been prospectively included in the database. Out of those, a total of 8050 (29.6%) patients with CLM had received preoperative systemic anticancer treatment before liver resection. In 2793 cases (34.7%) the disease was considered initially unresectable, and their data were complete and eligible for inclusion in the current study (Figure 1).

Irresectability was defined as the technical inability to completely remove all metastases while leaving at least 30% of normal liver parenchyma. Thus, in these patients, PO-SACT was administered with downsizing conversion intent.

### 2.2. Preoperative Management and Hepatic Resection

The oncologist in charge selected the most appropriate chemotherapy regimen after discussion in a multidisciplinary staff meeting according to local experience, tumor genotype, medical and biological condition and individual clinical treatment history, abiding by the most up-to-date international recommendations. Basically, all patients received fluoropyrimidines (leucovorin-modulated 5-fluorouracil or capecitabine), in association with irinotecan, oxaliplatin or both (triplet combination), and with or without targeted therapies [6].

The response to chemotherapy was evaluated every four to six cycles of treatment with computed tomography of the thorax, abdomen and pelvis according to the Response Evaluation Criteria in Solid Tumours Criteria [18]. Preoperative imaging often included contrast-enhanced hepatic Nuclear Magnetic Resonance imaging, 18F-FluoroDeoxyGlucose Positron-Emitting Tomography and/or hepatic ultrasound, as individually indicated.

During surgery, the consensual policy was to resect all the lesions detected at diagnosis, including remnant calcifications or scar lesions, with the aim to completely clear the liver of palpable and visible tumoral tissue, sparing the highest amount of liver parenchyma possible, as recommended [3,19].

### 2.3. Postoperative Chemotherapy

The indication of the regimen choice and the duration of postoperative therapy were decided by the lead oncologist following a multidisciplinary team discussion after surgery. In the absence of standardized guidelines [12], the most consensual approach was to aim at administering the same regimen used pre-operatively in case of confirmed radiological and pathological activity and satisfactory tolerance, as common practice in other peri-operative onco-surgical strategies [14,20,21,22,23].

### 2.4. Follow-Up

Patients were followed at 1 month postoperatively and then every 3 to 4 months with serum tumor markers (carcinoembryonic antigen [CEA] and CA 19.9), clinical examination and diagnostic imaging, as per local practice. Repeat resection of intra- and/or extrahepatic disease recurrence was performed when curative resection could be achieved. Further chemotherapy was administered if deemed indicated.

### 2.5. Statistical Analyses

The primary endpoint was overall survival (OS), calculated from the time of surgical resection till death or last follow-up visit. Secondary endpoints included: disease-free survival (DFS), calculated from the time of surgical resection till recurrence in any site or death, and hepatic-specific relapse-free survival (HS-RFS), calculated till the occurrence of liver recurrence only or death.

The main aim of the study was to identify the features of preoperative therapy associated with the longest OS after liver resection. In particular, we focused on the duration and type of PO-SACT. For the first aim, we categorized chemotherapy duration into short (6 cycles or less) or long (7 cycles or more) if the first-line regimen [20]. In the case of the second-line, the cut-off for short PO-SACT was set at less than 13 cumulative cycles, whereas for the third-line short PO-SACT was defined as comprising 18 or fewer cycles. Based on the approach used in prospective trials of onco-surgical management of liver metastases from colorectal cancer, where blocks of 6 cycles of chemotherapy were used [14,20,21,22,23], we considered that 7 or more cycles constituted a “long” chemotherapy duration. We selected the same cut-off of up to 6 cycles for each preceding chemotherapy line as short for consistency. For the type of PO-SACT, we focused on 4 specific, clinically-pertinent aspects [6]: impact of fluoropyrimidine choice (IV 5Fluorouracil versus PO capecitabine) [24]; impact of backbone cytotoxic agent (oxaliplatin versus irinotecan) [25]; impact of triplet combination [26]; impact of monoclonal antibodies (targeting EGFR or VEGF) [27].

We first determined OS probabilities as a function of different features of PO-SACT with the Kaplan-Meier method and compared them using the log-rank test. Then we identified the features of PO-SACT independently predicting for longer OS using multivariable Cox proportional hazard models, accounting for other relevant known potential clinical prognostic factors, including period of surgery (cutoff year: 2005) and Institution (continent-wise).

For the secondary endpoints (DFS and HS-RFS), we proceeded with the same approach.

Comparisons of baseline clinical-demographic characteristics between subgroups of patients defined by PO-SACT features were performed using the Chi^2^ test for categorical data and the independent-samples t-test or analyses of variance for continuous data.

The threshold for statistical significance was set for a *p* value ≤ 0.05. All statistical analyses were performed using SAS software version 9.1.3 (SAS Institute Inc., Cary, NC, USA).

## 3. Results

### 3.1. Study Population

At the time of LiverMetSurvey database analysis (December 2020), our study population included 2793 patients having received conversion PO-SACT for initially non-resectable disease (Figure 1). The main clinical-demographic characteristics of the study population are summarized in Table 1.

### 3.2. Preoperative Chemotherapy Features and Outcomes

Table 1 details the main features of pre-operative systemic anticancer treatment administered with conversion intent prior to surgical resection.

Altogether, the most frequent PO-SACT regimen used was FOLFOX: in 28.3% of patients, it was used alone, and with bevacizumab or cetuximab/panitumumab in further 14.2% and 10.4% of patients, respectively. In comparison, FOLFIRI was used alone, in combination with anti-VEGF or with anti-EGFR monoclonal antibodies (mAb) in 10.8%, 10.7% and 8.6%, respectively. Over a third of the patients received less than 7 pre-operative cycles, and resection after 3rd or more line therapy was rare.

Postoperative chemotherapy was administered to slightly less than half of the patients in this setting.

### 3.3. Surgical Complications

Table 1 presents the details of postoperative morbidity and mortality. For most of the patients, the postoperative course was uneventful, with an average duration of hospitalization of less than a fortnight; however, some histological abnormalities in the non-tumoral hepatic parenchyma were observed in almost half of the patients as a likely consequence of PO-SACT (Table 1).

### 3.4. Pre-Operative Systemic Anticancer Therapy Duration

Shorter (as defined above) PO-SACT duration was associated with significantly better OS (median: 41.0 [38.1–44.7] and 36.5 [32.9–40.1] months; *p* = 0.021; Figure 2a). Altogether, clinical-demographic features were comparable with regards to conversion PO-SACT duration (data not shown). Notwithstanding, imbalanced clinical characteristics included higher proportion of synchronous metastases at diagnosis (by 6.2%) and greater degree of liver involvement (in terms of number: 7.3% more patients with at least 4 lesions, and size: 6.4% more patients with the largest lesion greater than 3 cm) in the subgroup with longer PO-SACT (*p* < 0.05).

For all secondary endpoints, too, shorter PO-SACT was associated with better outcomes: *p* = 0.006 for DFS and *p* = 0.024 for HS-RFS (Figure 2b,c).

### 3.5. Pre-Operative Systemic Anticancer Therapy Modalities

No PO-SACT modality explored was associated with OS differences in patients with initially non-resectable liver disease. Thus, oxaliplatin- or irinotecan-containing regimens yielded comparable outcomes (respective median OS: 39.7 [36.0–42.3] and 38.0 [34.9–43.0] months; *p* = 0.64; Figure 3a), and so did fluoropyrimidine type (38.6 [36.0–41.4] for IV and 40.2 [35.5–51.1] for PO; *p* = 0.47; Figure 3b), triplet (36.1 [26.6–46.7]) and doublet (39.0 [36.9–41.8]) regimens (*p* = 0.59) and the addition or not of targeted agents (anti-VEGF: 37.7 [34.9–41.9]; anti-EGFR: 42.5 [36.6–50.0]; none: 38.2 [34.9–41.8]; *p* = 0.12; Figure 3c). Nevertheless, the cohort treated with cetuximab or panitumumab appeared to have slightly better outcomes. The main differences in the clinical characteristics of patients according to different treatment modalities included more common synchronous metastases in patients treated with irinotecan, bulkier liver involvement in patients treated with triplet combination or mAbs, and older age in patients having received capecitabine (data not shown).

DFS, however, was marginally longer with doublet (median: 15.2) than with triplet regimen (12.0; *p* = 0.048), whereas all the other comparisons yielded similar outcomes (not shown).

For HS-RFS, too, there was a trend towards better outcomes on doublet therapy (*p* = 0.055). Additionally, HS-RFS was significantly longer without mAbs (median unattained) than with anti-EGFR (33.0) or anti-VEGF (33.2; *p* = 0.001); this secondary outcome, too, did not seem to be influenced by choice of the backbone agent or of the fluoropyrimidine (not shown).

### 3.6. Multivariable Analyses

The final multivariable model included 1713 patients (61.3%).

Longer (i.e., more than 6 cycles in first-line, or more than 12 total cycles in second-line) PO-SACT duration was independently confirmed as a prognostic factor of poorer OS (HR: 1.18; *p* = 0.046); moreover, at multivariable analysis, the use of no targeted agent or of anti-VEGF mAb was retained as a predictor of shorter OS (HR: 1.24; *p* = 0.0491). The other independent negative prognostic factors are shown in Table 2. Primary tumor localization had no impact on any outcome (data not shown).

Longer PO-SACT was also an independent predictor of shorter DFS (HR: 1.23; *p* = 0.016) and HS-RFS (HR: 1.25; *p* = 0.0512).

## 4. Discussion

In this large international registry-based analysis, we identified short (<7 cycles in first-line and <13 cycles in second-line) pre-operative systemic anticancer therapy duration as associated with longer OS, DFS and HS-RFS after surgery of initially-irresectable hepatic metastases (Figure 2; Table 2); this result remained independent from other known clinical prognostic factors or treatment (chemotherapy or surgery) features, for all outcomes.

Thus, in initially unresectable disease, a longer duration of PO-SACT could be assumed as a surrogate of insufficient down-sizing capability due to an extensive tumor load or to a chemoresistant biological tumoral phenotype. Additionally, prolonged pre-operative treatment could increase the risk of disappearing metastases, which might not have been completely sterilized by the PO-SACT [28,29]. Thus, our findings support performing liver surgery as soon as technically feasible after active PO-SACT; this approach of early resection could also reduce the risks of chemotherapy-induced hepatotoxicity [30,31], therefore decreasing postoperative complication risks; this aspect could be particularly beneficial for elderly patients, altogether at higher risk of postoperative mortality after liver resection [32]; moreover, shorter chemotherapy exposure would also arguably reduce the incidence of other cumulative toxicities, thus allowing potential re-challenge with active anticancer medications, postoperatively or in case of relapse.

In accordance with observed outcomes in metastatic disease independently of any surgical strategy [24,33,34], oral fluoropyrimidine yielded similar survival results to the infusional one in the pre-operative conversion setting (Figure 3a). Thus, our findings provide reassurance to medical oncologists in their choice of fluoropyrimidine with the initially not-resectable disease to continue being based on toxicity profiles, practicalities and patient’s choice [24], allegedly without hindering long-term outcomes of the onco-surgical strategy. Nonetheless, 5FU remained more frequently used than capecitabine in our international registry, in accordance with multicentric experience in the adjuvant setting [35].

In accordance with expected antitumor activity being highly comparable between oxaliplatin- and irinotecan-based chemotherapy [36]. We did not find any significant difference in OS, PFS and HS-RFS according to the cytotoxic treatment backbone, even at univariable analysis (Figure 3b). Indeed, in the non-resectable setting, there is no robust evidence to prefer oxaliplatin or irinotecan [5,36,37]. Thus, these findings also back the choice of the most appropriate doublet regimen for each individual patient, without concerns of negatively impacting on outcomes.

Somewhat unexpectedly, in our surgical database we only observed independent survival prolongation with intensified conversion PO-SACT, from the adjunction of either cetuximab or panitumumab (Figure 3c), whereas neither triplet regimen nor the association of bevacizumab showed survival advantage; this is even more surprising considering that most of the patients in this cohort having received anti-EGFR treatment had been treated before tumor RAS genotyping became standard practice [38]; thus, a sizeable proportion (expectedly, 34–54%) [39] of them might have RAS mutated tumors with no anticipated benefit from targeting EGFR, diluting the positive effect on RAS-wild type cases. Notwithstanding, it must be acknowledged that the prognostic impact of RAS mutation [40,41] might have amplified the survival in favour of anti-EGFR mAbs, more recently used in the appropriate genotype-selected subgroup. Additionally, the apparent benefit conferred by anti-EGFR targeting on OS was not observed for DFS and HS-RFS: indeed, for both these secondary outcomes, better results were found in the patient not having receiver either mAb. Notably, despite both cetuximab and panitumumab added to doublet regimens in first- or second-line have shown to be superior to placebo in metastatic disease in terms of overall survival [42,43], they have failed to improve outcomes in the adjuvant setting [44], and cetuximab also in the peri-operative setting of resectable hepatic metastases [14]. Notwithstanding, independently from an onco-surgical strategy, targeting EGFR appears to be superior to targeting VEGF in metastatic (RAS not-mutated) disease [45].

Altogether, since intensive conversion PO-SACT, including triplet with biologicals or hepatic artery infusion, is associated with high response and secondary resection rates [46,47], even when used as salvage options in pre-treated patients [26,48,49], it could be argued that if an intensive regimen is the only valid therapeutic option in selected patients with unresectable liver-predominantly metastases, the choice of the triplet, a biological, hepatic artery infusion or a combination of them within an aggressive onco-surgical strategy can still be expected to be associated with overall better survival chances than with a bland palliative chemotherapy, even in second-line [16,50,51]. Thus, the more effective is the conversion chemotherapy, the higher is the chance of inducing resectability of previously unresectable patients [4,5]. Indeed, a significant positive correlation has been demonstrated between objective response rate and secondary liver resection rate [49]. Altogether, our real-world findings seem to suggest that, as long as the pre-operative regimen is effective in downsizing the liver disease and allow resection, the combination of drugs is of marginal impact on patient outcomes.

This introduces what we believe to be the main limitation of the current study: the intrinsic nature of our heterogeneous surgical database does not consider the patients referred to the oncologist for optimal chemotherapy for metastatic colorectal with hepatic involvement and never resected. Hence, we are unable to gauge with this “surgical” registry to which extent each type of conversion PO-SACT regimen was able to downsize the hepatic disease enough to allow secondary liver resection. Additionally, we did not routinely collect information about molecular subtypes classification, individual mutational genotyping, or immune-microenvironment, all of which play a role in prognosis and response to medical therapy [6,52,53,54,55]. Finally, given the follow-up required to obtain meaningful outcome information, we do not dispose as yet of data about the role of novel therapies, such as immune checkpoint inhibitors [56], in long-term outcomes after liver resection.

Nevertheless, all the patients in this cohort were discussed in multidisciplinary team meetings involving both oncological and surgical expertise, as recommended [20]; moreover, the large, international pool of practices feeding into the LiverMetSurvey registry permits a certain degree of confidence in generalizing the findings in terms of prognostic PO-SACT determinants. Thus, the other prognostic factors evidenced in the study (Table 2) are well established [57,58,59], further confirming the extendibility of the results from our cohorts of patients treated in daily practice. Similarly, overall, the results were fairly robust in terms of outcomes, with a high degree of overlap among OS, DFS and HF-RFS.

## 5. Conclusions

In summary, our real-world data in a large international prospective registry revealed that best OS, DFS and HS-RFS are achieved when resection is performed after 6 or fewer cycles of pre-operative first-line treatment (12 for second-line) in patients with the non-resectable disease. Furthermore, our study did not identify the best pre-operative regimens, but suggested that, whenever indicated, the addition of anti-EGFR mAbs to the cytotoxic backbone ought to be preferred. Therefore, optimally active chemotherapy for the individual patient and a limited number of cycles should be aimed towards in as much as the main objective of the strategy. We recommend that reviewing each patient every 2 months at a multidisciplinary meeting as recently proposed by an expert panel is the best means to achieve this objective.

## Figures and Tables

**Figure 1 cancers-14-04340-f001:**
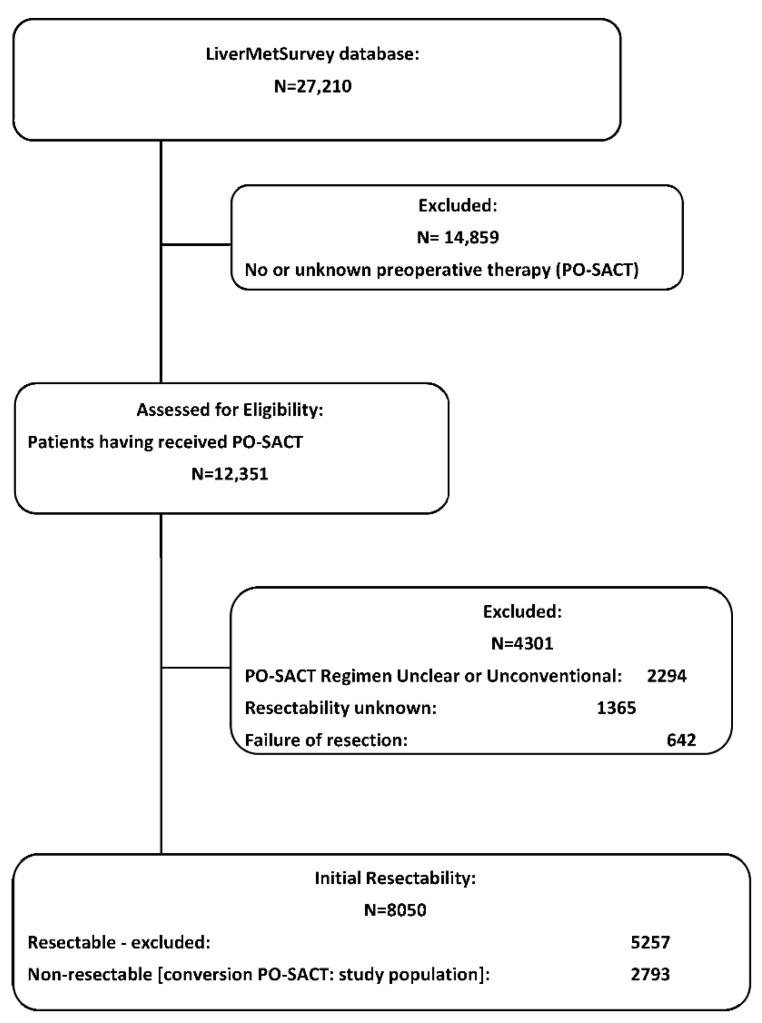
Study Flowchart (CONSORT diagram). PO-SACT: pre-operative systemic anticancer therapy; CONSORT: consolidated standards of reporting trials.

**Figure 2 cancers-14-04340-f002:**
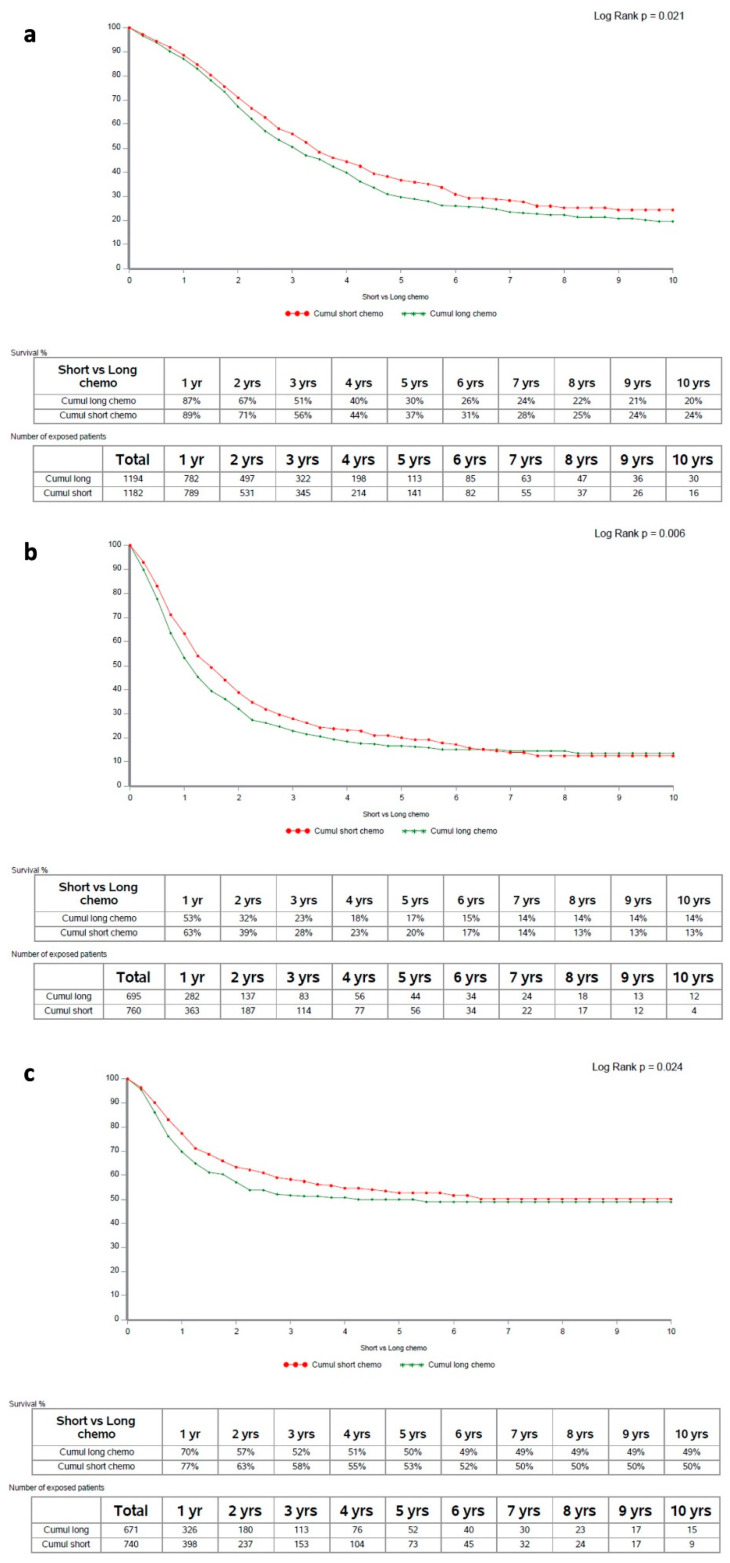
Survival curves according to PO-SACT duration: panel (**a**) OS; panel (**b**) DFS; panel (**c**) HS-RFS. PO-SACT: pre-operative systemic anticancer therapy; OS: overall survival; DFS: disease-free survival; HS-RFS: hepatic-specific relapse-free survival.

**Figure 3 cancers-14-04340-f003:**
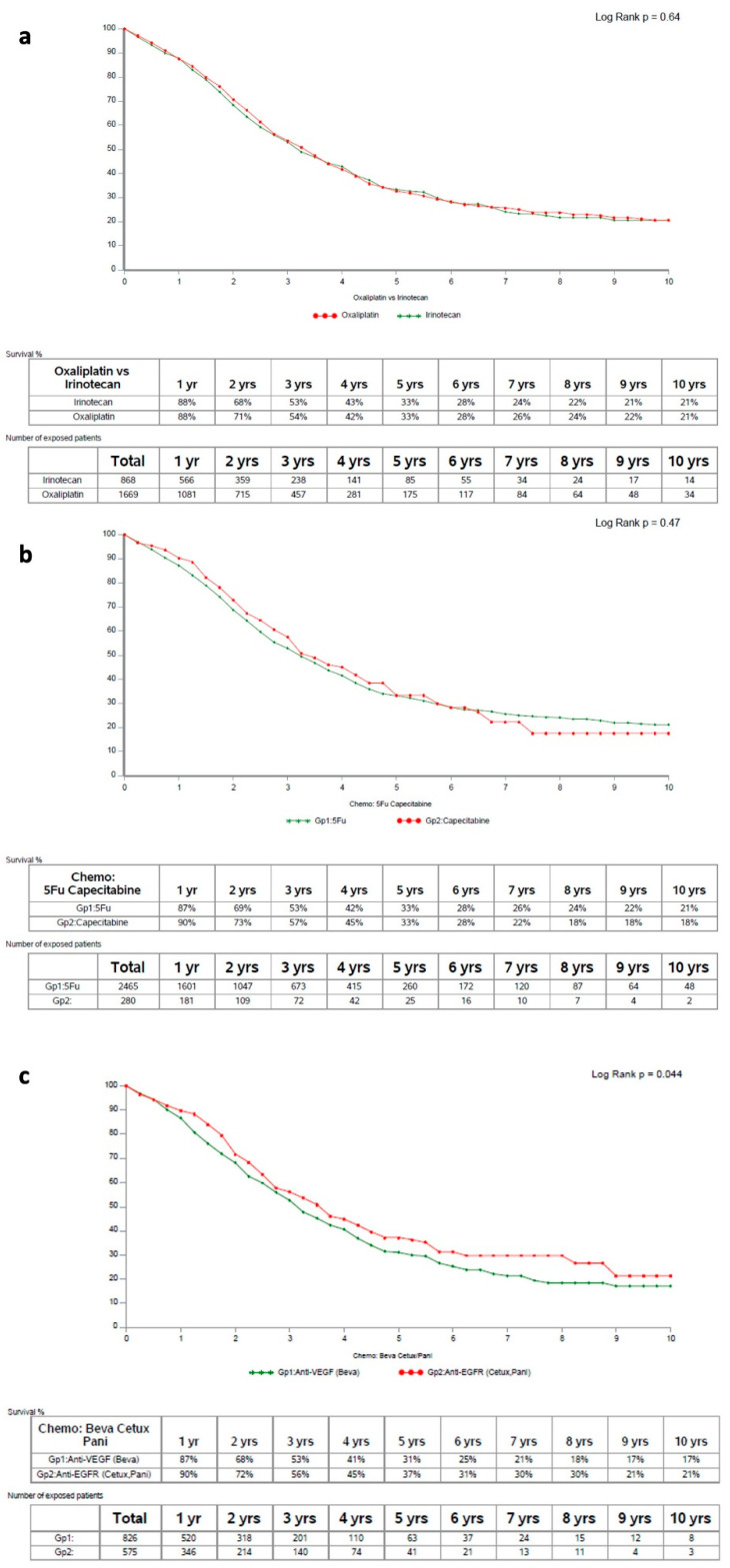
Overall Survival curves according to considered PO-SACT modalities: panel (**a**): oxaliplatin versus irinotecan; panel (**b**): 5-fluorouracil versus capecitabine; panel (**c**): cetuximab/panitumumab versus bevacizumab. PO-SACT: pre-operative systemic anticancer therapy.

**Table 1 cancers-14-04340-t001:** Main demographical, clinical and surgical characteristics of the study population (N = 2793) and of the PO-SACT features and outcomes.

Characteristic	(N = 2793)
Age [median; IQR] Mean ± SD	[61.3; 14.8] 60.6 ± 12.3
Gender [M/F]; N (%)	1712 (61.4)/1076 (38.6)
Primary Tumor Location [Left-sided/Right-sided/multiple]	2131 (77.6)/549 (20.0)/65 (2.4)
Adjuvant chemotherapy [Y]	738 (28.5)
Neoadjuvant radiotherapy [Y]	270 (10.4)
Synchronous Liver Metastases [Y]	2168 (78.0)
Number of liver metastases [1/>3/>7]	440 (18.4)/1341 (56.2)/611 (25.6)
The largest size (mm) of liver metastases (mean ± SD)	48.7 ± 42.4
At least one lesion ≥50 mm [Y]	929 (40.1)
Bilobar involvement [Y]	1984 (71.9)
Concomitant extrahepatic disease [Y]	305 (12.9)
Type of Resection [non anatomical/staged/combined techniques/portal occlusion]	1826 (67.5)/614 (22.4)/1259 (46.1)/734 (27.1)
Major hepatectomy [Y]	1910 (70.3)
Only 1 hepatectomy/patient [Y]	2073 (74.2)
Microscopically complete resection [Y]	1622 (70.7)
Complete pathological response [Y]	244 (10.2)
Length (days) of post-op hospital stay (mean/SD)	13.0/10.4
Postoperative complications [Y]	891 (34.6)
Abnormal non-tumoral liver [Y]	1379 (59.9)
1-month mortality [Y]	38 (1.5)
**PO-SACT Feature**	**N (%)**
**Regimen drugs [Y]**	
5-FluoroUracil	2465 (88.3)
Capecitabine	280 (10.0)
Oxaliplatin	1669 (59.8)
Irinotecan	868 (31.1)
Anti-EGFR mAb	575 (20.6)
Anti-VEGF mAb	826 (29.6)
Triplet combination	208 (7.4)
Doublet	2537 (90.8)
**Line of treatment**	
1	2333 (83.5)
2	333 (11.9)
3+	127 (4.5)
**Cumulative # of cycles**	
(mean/SD)	7.5/4.0
Median	8.0
IQR	[6; 12]
1–6	1046 (37.5)
7–12	1003 (35.9)
13+	369 (13.2)
UK	375 (13.4)
**Objective response to PO-SACT**	
CR	70 (2.5)
PR	2103 (75.2)
NC/SD	359 (12.9)
PD	109 (3.9)
NA or NE	152 (5.4)
**Received postoperative chemotherapy [Y]**	
	1341 (48.0)

CR: complete response; EGFR: epithelial growth factor receptor; IQR: interquartile range; mAb: monoclonal antibody; N: number; NA: not available; NC/SD: no change/stable disease; NE: not evaluable; PD: progression of disease; PO-SACT: pre-operative systemic anticancer therapy; PR: partial response; SD: standard deviation; UK: unknown; VEGF: vascular endothelial growth factor; Y: yes.

**Table 2 cancers-14-04340-t002:** Multivariable proportional hazard Cox models for overall survival.

Parameter	Value	Univariate *p*	Multivariable *p*	Multivariable HR	HR 95% Confidence Limits
Liver curative surgery	No	<0.0001	<0.0001	1.77	1.43; 2.17
Concomitant extrahepatic disease	Yes	0.003	0.0442	1.27	1.01; 1.61
Isolated liver metastasis	No	<0.0001	0.0022	1.44	1.14; 1.82
Nodal involvement of primary	Yes	<0.0001	0.0051	1.28	1.08; 1.53
First-line PO-SACT	No	0.0008	0.0003	1.50	1.21; 1.87
Use of anti-EGFR mAb	No	0.081	0.0491	1.24	1.00; 1.55
PO-SACT duration	Long	0.001	0.0460	1.18	1.00; 1.39

EGFR: epithelial growth factor receptor; HR: hazard ratio; mAb: monoclonal antibody; PO-SACT: pre-operative systemic anticancer therapy.

## Data Availability

Specific parts of the LiverMetSurvey dataset used for the current manuscript could be provided in anonymized form upon request, pending approval by the LiverMetSurvey scientific committee, for explicit queries.

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
