# Peer review of "Impact of Preoperative Chemotherapy Features on Patient Outcomes after Hepatectomy for Initially Unresectable Colorectal Cancer Liver Metastases: A LiverMetSurvey Analysis"

_cancers, 2022, doi:10.3390/cancers14174340_

Round 1

Reviewer 1 Report

The work is extremely valuable. Collected data from various centers include 27 210 patients, in years 1995-2020. A lot of data was included in the analysis, hence it is extremely difficult to summarize. The Authors referred in the discussion to the table 3(line 319), which I did not find in the paper. The table was supposed to relate to the predictive factors analyzed. A very interesting discussion and careful conclusion emphasise the value of the work. The references has been well selected.

Author Response

We sincerely thank the Reviewer for their comments.

We have corrected the wrong reference to Table 3 with Table 2 instead.

Reviewer 2 Report

This is a very interesting, well written study with a concise conclusion. 

As mentioned in the limitations, it suffers from its database characteristics but nevertheless provides information on an international, complex patient cohort. 

Major points:

The study cohort is very heterogenous - there are few informations on i.e. primary tumor characteristics and therapy as well as postoperative chemotherapy (while administered in 48%!), that makes conclusions  hard to draw

Did you assess whether primary tumor localisation (left/right/multiple) had an impact on outcome?

Minor points:

Study flowchart is a bit misleading - the true study population of 2,793 is printed very small in the end, text and symbols dont fit well

Add an overview of the abbreviations and consider including them in table / figure titles

choose either PC-SACT or P-SACT as abbreviation, otherwise this might lead to confusions

Author Response

We agree with the Reviewer about their comments.

We indeed assessed the impact of primary tumor localisation and it had no impact on survival (p>0.093). We added a sentence in the results section to report this finding unequivocally.

We re-edited Figure 1, and increased and put in bold the final study population.

We added a general list of abbreviations at the end of the text, as well as a specific list in each figure and table legends.

We apologize for the confusion created by the misuse of two abbreviations for the same concept: we have now solely and consistently used PO-SACT to refer to pre-operative systemic anticancer therapy throughout our manuscript.

Reviewer 3 Report

It is suggested to compare whether there are significant differences in patient characteristics between the short P-SACT duration group and the long P-SACT duration group, and to perform a univariate analysis for the patients.

Author Response

We thank the Reviewer for their comments. We had indeed provided brief description of the (indeed minimal) statistically-significant differences in clinical-demographic features between the subgroups with short and long PO-SACT in the results section (paragraph: pre-operative chemotherapy duration). We agree nevertheless about the relevance of this point, and we have added some text to report more details on the significant differences.

We performed the univariate analysis on OS and the parameters showed in Table 2 were all significantly associated with survival (p<0.1). We decided not to report all Hazard ratios and their confidence limits to avoid confusion in the table, but we added a column to report the univariate p values for completeness.